# Organic Acid Supplementation in Worker Honeybees (*Apis mellifera*): Impacts on Glandular Physiology and Colony Resilience

**DOI:** 10.3390/insects16121203

**Published:** 2025-11-26

**Authors:** Gebreamlak Bezabih, Tesfay Atsbha, Solomon Zewdu Altaye, Qingsong Zhou, Jianke Li, Christian W. W. Pirk, Chaodong Zhu, Yu Fang

**Affiliations:** 1Tigray Agricultural Research Institute, Mekelle P.O. Box 492, Ethiopia; atsbhatesfay@gmail.com; 2Ethiopian Institute of Agricultural Research, Chiro P.O. Box 190, Ethiopia; szaltaye@gmail.com; 3Key Laboratory of Zoological Systematics and Evolution, Institute of Zoology, Chinese Academy of Sciences, Beijing 100101, China; zhouqingsong@ioz.ac.cn (Q.Z.); zhucd@ioz.ac.cn (C.Z.); 4State Key Laboratory of Resource Insects, Institute of Apicultural Research, Chinese Academy of Agricultural Sciences, Beijing 100193, China; apislijk@126.com; 5Social Insect Research Group (SIRG), Department of Zoology & Entomology, University of Pretoria, Private Bag X20, Hatfield, Pretoria 0028, South Africa; christian.pirk@up.ac.za

**Keywords:** *Apis mellifera*, hypopharyngeal gland, organic acids, citric acid, royal jelly, glandular development, gut microbiota, detoxification, colony resilience, sustainable apiculture

## Abstract

**Simple Summary:**

Honeybees play a vital role in pollinating crops and preserving healthy ecosystems, but their survival is often threatened by poor nutrition, exposure to pesticides, and disease. This review examines the ecological and physiological functions of organic acids in the honeybee diet, with a particular emphasis on how they affect glandular development and colony resilience. Organic acids, including citric, lactic, and acetic acid, and substances derived from plant sources, including p-coumaric and indole-3-acetic acid, influence the functionality of the hypopharyngeal, mandibular, and wax glands of honeybees. They do so through metabolic activation, microbiota modulation, and detoxification. Organic acids also promote gut health and increase resistance to stressors of the environment, while enhancing immune function and wax secretion, as well as the quality of royal jelly. Organic acids provide alternatives to synthetic treatment options with low residue and align with sustainable beekeeping practices. This review raises the importance of undertaking integrative studies that incorporate ecological, behavioral, and molecular methods to demonstrate long-term benefits and develop adaptive feeding strategies.

**Abstract:**

Honeybees require diverse nutrients for larval growth, adult development, and colony health. Pollen quality significantly impacts reproduction, productivity, and growth. Bioactive substances from honeybee glands enhance colony health, with recent studies showing that optimal citric acid intake extends lifespan, boosts pollen consumption, accelerates mandibular gland development, and improves royal jelly quality. This review examines organic acid feeding’s effects on gland development and overall health, offering insights for beekeeping and supplementary food development to support sustainable apiculture. Research gaps in organic acid supplementation, gland development, and health benefits are identified. The impact of varying organic acid concentrations on 10-HDA biosynthesis in mandibular glands and key regulatory proteins influencing 10-HDA expression is summarized. Findings highlight the benefits of organic acid supplements for worker bee gland development and health, guiding future research and practical applications in beekeeping.

## 1. Introduction

Honeybee glands are central to colony function, contributing to communication, wax synthesis, and food processing [1]. Specialized glands, including the mandibular, hypopharyngeal, wax, and poison glands, are anatomically distributed and undergo age-dependent development and functional shifts [2,3,4,5,6]. For instance, wax glands located ventrally on the abdominal segments of nurse bees secrete wax for nest construction [7,8], while the mandibular and Nasonov glands contribute to pheromonal communication [9,10,11,12,13,14]. Among these, the hypopharyngeal glands (HPGs) are pivotal in synthesizing royal jelly, key enzymes such as invertase and oxidase, and nutrient-rich secretions for brood and adult bees [15,16]. Gland size and activity are highly responsive to dietary inputs, especially the quantity and quality of pollen and its substitutes [15,16,17,18,19,20,21,22]. Building on foundational anatomical and behavioral studies by Winston [1] and Seeley [5], this review explores how nutritional inputs, particularly organic acids, modulate gland development and colony-level resilience.

The mandibular glands (MDGs) secrete 10-hydroxy-2-decenoic acid (10-HDA), a key component of royal jelly, which is critical for queen development and colony health [23,24]. Glandular output is closely tied to nutritional intake, particularly pollen, which supplies proteins, lipids, vitamins, and minerals essential for biosynthesis [25]. Nurse bees digest pollen to synthesize royal jelly, which is fed to all larvae during their first three days. Larvae destined to become workers or drones then change their diet to worker jelly (a mixture of honey and pollen), while queen larvae continue receiving royal jelly throughout development. However, pollen scarcity or nutritional imbalance, often driven by habitat degradation or seasonal shifts, can impair gland development and reduce colony productivity [17,26,27,28,29].

Emerging evidence indicates that organic acids, such as citric, acetic, and lactic acids, can enhance gland development and buffer against nutritional stress [30]. These compounds have been shown to improve survival under pesticide exposure [31], stimulate detoxification pathways [32], and promote HPG growth [33]. This review synthesizes current findings on organic acid supplementation and its physiological impacts on honeybee gland development, colony health, and resilience, offering insights for sustainable apiculture and future research.

## 2. Nutritional Foundations and Pollen-Based Diets

Pollen serves as the nutritional cornerstone for honeybees, supplying essential macronutrients and micronutrients, including proteins, lipids, vitamins, and minerals, that underpin gland development and immune function [1,34]. Reviews by Nicolson et al. [35] and Frias et al. [36] underscore the biochemical variability and digestibility of pollen across floral sources, seasons, and regions, framing its central role in honeybee physiology. Comparative studies in *Apis cerana* reveal species-specific responses to pollen types and substitutes, with oilseed rape pollen supporting optimal HPG development and midgut enzyme activity [37]. These findings highlight the need for tailored nutritional strategies across honeybee species. The synthesis of royal jelly is tightly linked to the availability and biochemical quality of pollen, particularly its amino acid profile and phenolic content [17,19]. Pollen composition exhibits substantial variability across floral sources, seasons, and geographic regions, influencing its nutritional value and digestibility [26,34,36,38]. For instance, a honeybee colony requires a diet containing approximately 20–25% protein for its proper growth and survival [38]. Reported protein content in angiosperm pollen varies widely, from as low as 2.5% to over 60% of dry weight, depending on floral source and season [35,36]. Pollen scarcity or suboptimal nutritional profiles are associated with reduced worker longevity, impaired glandular development, and compromised colony performance [17,22,27,39,40]. Moreover, environmental stressors such as pesticides, pathogens, and climate variability exacerbate nutritional deficits, further compromising bee health [27,41].

To address these challenges, researchers have explored supplementary feeding strategies, including pollen substitutes and bioactive compounds [42]. Organic acids, including citric, lactic, and acetic acids, represent a biologically grounded strategy for supplementing pollen-based diets, with emerging evidence supporting their role in enhancing gland function and metabolic resilience [30,43]. These acids may buffer nutritional stress, promote glandular activation, and support colony-level resilience under environmental and dietary constraints [32,33,43,44].

## 3. Gland Biology and Age Polyethism

In honeybees, worker roles shift as they age, a process known as age polyethism. Gland development in honeybees is tightly coupled with age polyethism, the age-based division of labor that governs physiological transitions from nursing to foraging roles [16,45,46,47,48]. Seeley’s synthesis of age polyethism [5] provides a conceptual scaffold for understanding how glandular remodeling aligns with behavioral shifts and colony demands. For example, the HPGs, which are heavily involved in producing royal jelly during the nursing phase, diminish in activity as bees prepare for foraging. Simultaneously, other glands develop further to enhance pheromone production necessary for communication among adults. These transitions are orchestrated by hormonal signals, nutritional inputs, and environmental cues that collectively shape gland morphology and activity [3,4,5,42].

HPGs, located in the worker bee’s head, are among the most developmentally plastic, responding rapidly to age and nutritional status [2,13,49]. HPGs initiate development post-emergence and reach peak activity during the nursing phase, when workers produce royal jelly for brood and queen nourishment [2,49]. Acinar diameter expands markedly during this phase, reflecting elevated protein synthesis and secretory output [15,17]. As workers age into foragers, HPGs regress and secretory activity declines, mirroring shifts in behavioral roles [48,50]. This plasticity enables colonies to dynamically adjust glandular output in response to labor demands and nutritional constraints [16,48,51,52].

MDGs also undergo age-dependent remodeling [45,46,47]. In nurse bees, MDGs secrete royal jelly lipids such as 10-HDA, critical for larval development and queen physiology [23,24]. With age, MDG output shifts toward pheromonal compounds that mediate social signaling and colony defense [47,53]. This ontogeny reflects the colony’s adaptive balance between brood care, social coordination, and environmental responsiveness. Wax glands, situated bilaterally in the abdomen, follow a similar age-dependent trajectory [7]. Peak wax gland activity occurs between days 10–18, aligning with nest construction and comb building [7,54]. Wax gland cells transition from cuboidal to elongated secretory forms, reflecting developmental stage and task allocation [8,55]. Following peak secretion, wax glands regress as workers transition to foraging roles [8,54].

The dynamic nature of gland development underscores the need for targeted nutritional support during key life stages. Organic acids, by enhancing protein metabolism and glandular activation, may sustain optimal gland function under environmental and dietary stress. This is especially relevant in degraded landscapes or pollen-scarce seasons, where supplementation can buffer against physiological decline and support colony function.

## 4. Supplementary Feeding Practices

Supplementary feeding remains a cornerstone of modern apiculture, particularly in regions or seasons where natural forage is scarce or nutritionally deficient. To support colony development and mitigate stressors, including disease, pesticide exposure, and climate variability, beekeepers employ diverse feeding strategies aimed at enhancing productivity and resilience. Paray et al. [56] and Pudasaini et al. [57] have reviewed the evolution of supplementary feeding strategies, highlighting both the promise and limitations of artificial diets in sustaining colony health. Common supplements include sugar syrups, artificial pollen substitutes, and protein-rich cakes formulated to mimic the nutritional profile of natural pollen [20,33]. These are often formulated with ingredients such as honey, sucrose, soybean flour, brewer’s yeast, and vitamins to mimic the nutritional profile of natural pollen [20,56,57]. Electrolyte solutions, often enriched with minerals, are occasionally administered to support thermoregulation, hydration, and gut integrity, especially under heat stress or forage scarcity [58,59,60].

Organic acids are increasingly incorporated into supplementary feeds for their multifunctional benefits, including glandular activation, microbial modulation, and detoxification support [23,61,62,63]. Citric and lactic acids, when added to pollen substitutes or protein-rich diets, enhance protein metabolism, stimulate glandular growth, and improve royal jelly biosynthesis [23,44]. Acetic acid contributes to hive hygiene and gut health by lowering pH, inhibiting pathogens, and supporting probiotic symbiosis, particularly with *Lactobacillus rhamnosus* [33,64]. Essential oils, such as rosemary, thyme, mint, and clove, are also used as feed additives for their antimicrobial and immunomodulatory effects [56]. However, their application requires careful dosing and monitoring to avoid adverse effects on bee behavior and colony dynamics [65].

Seasonal timing is pivotal for effective supplementation, as nutritional demands shift with brood cycles, foraging intensity, and climatic stressors [40,66]. During early spring, protein-rich feeds support brood rearing and gland activation, while carbohydrate sources sustain foraging and thermoregulation [40]. In late summer and fall, supplements may help prepare colonies for overwintering by boosting fat body reserves and immune function [39,65]. Effective feeding regimes must account for colony-specific needs, forage availability, and regional stressors, including pathogen prevalence and landscape composition [67]. Monitoring colony response is essential. Indicators such as brood area expansion, gland morphology, foraging activity, and disease prevalence can guide adjustments in feed composition and frequency [40]. Adaptive feeding strategies, particularly those incorporating organic acids and engineered sterol supplements, offer a biologically grounded approach to enhancing colony resilience and productivity [68]. Economic analyses by Sultana et al. [69] demonstrate that low-cost supplements such as pumpkin syrup can reduce sugar syrup costs by up to 50%, while enhancing brood, honey, and pollen cell production. These findings reinforce the practical relevance of biologically grounded feeding strategies.

## 5. Organic Acids in Honeybee Nutrition

Building on the nutritional strategies outlined above, this section explores the specific physiological roles of organic acids in honeybee nutrition. Organic acids are increasingly recognized as multifunctional dietary additives in honeybee nutrition, with growing evidence supporting their roles in gland development, microbial modulation, and detoxification [23,31,70]. Ricigliano and Anderson [30] and Maggi and Mitton [43] have synthesized emerging evidence on the multifunctionality of organic acids, linking their roles to glandular activation, microbial modulation, and stress resilience. Naturally occurring in nectar, pollen, and microbial metabolites, compounds such as citric, lactic, acetic, tartaric, and p-coumaric acids contribute to bee physiology through metabolic, microbial, and stress-response pathways [23,31,60]. Their supplementation has shown promising effects on glandular growth, protein synthesis, and digestive efficiency, particularly under nutritional stress or pesticide exposure [23,31].

Citric acid, at concentrations of 0.50–0.75%, enhances the size and biosynthetic activity of HPGs, MDGs, and cephalic salivary glands. These changes support royal jelly production and wax secretion [23,71,72]. Citric acid supplementation has been shown to elevate 10-HDA and protein concentrations in royal jelly, while modulating its moisture and sugar profile [23,43]. These effects are likely mediated by citric acid’s role in enhancing energy metabolism and secretory activity in HPGs and MDGs [23,43]. These anatomical enhancements correlate with increased pollen consumption and extended worker lifespan, indicating improved nutrient utilization and colony productivity [23,73].

Lactic and acetic acids similarly increase acinar surface area in HPGs, improving enzyme synthesis and secretory capacity essential for royal jelly production [33,64]. When paired with probiotics such as *Lactobacillus brevis*, lactic acid further enhances royal jelly yield and glandular development [74]. Morphometric analyses by Hassan and Elenany [74] show that combining probiotics with soybean patties significantly increases HPG diameter and surface area, correlating with enhanced royal jelly output. These findings support the synergistic potential of probiotic-acid diets in nurse bee physiology. Acetic acid also supports wax gland activity and digestion, primarily through its role in gut pH regulation and symbiosis with acetic acid bacteria such as *Bombella* spp. [75,76,77,78]. Table 1 summarizes these effects by mapping each organic acid to its target gland(s) and associated physiological outcomes. For example, p-coumaric acid activates detoxification pathways via cytochrome P450 upregulation, improving survival under pesticide exposure [31,32]. Indole-3-acetic acid similarly enhances resilience to acaricide stress, suggesting a broader role for plant-derived acids in stress adaptation [31]. Beyond direct glandular effects, organic acids contribute to a healthier gut microbiome, facilitating nutrient metabolism and pathogen inhibition. Citric and acetic acids promote colonization by beneficial bacteria such as *Snodgrassella alvi*, which aid in digestion and immunity [70,75,79]. Lactic acid, especially when paired with *Lactobacillus rhamnosus*, enhances HPG growth and reduces pathogen loads, including *Nosema ceranae* [33,64]. These microbial interactions indirectly support glandular function by improving overall physiological health and resource assimilation [60]. Together, these compounds exert synergistic effects across glandular physiology, microbial symbiosis, and detoxification. While this section focuses on anatomical and functional outcomes, the broader implications for honeybee health, including immunity, resilience, and colony-level vitality, are explored in Section 7.

## 6. Mechanisms of Action and Molecular Insights

Organic acids exert their physiological effects through a constellation of molecular pathways involving gene expression, proteomic activation, and microbial symbiosis. Transcriptomic and proteomic studies reveal that compounds such as citric, lactic, and acetic acids stimulate protein synthesis in HPGs and MDGs, enhancing the secretion of royal jelly and other bioactive substances [2,72]. Proteomic insights from Hu et al. [2] and transcriptomic analyses by Ueno et al. [16] reveal that organic acids influence gene expression and energy metabolism in HPGs and MDGs. As a central metabolite in the TCA cycle, citric acid facilitates lipid biosynthesis and energy production within glandular tissues, thereby enhancing royal jelly output [23,43].

These acids also modulate nutrient assimilation and metabolic signaling. Citric and lactic acids upregulate genes involved in sugar and lipid metabolism, while acetic acid enhances digestive efficiency and microbial homeostasis via its symbiosis with *Bombella* spp. and *Snodgrassella alvi* [30,70,75]. Proteomic analyses confirm that organic acid supplementation primes glandular tissues for active secretion by activating energy metabolism and cellular proliferation [2,72]. These relationships are synthesized in Figure 1, which maps organic acid types to their physiological mechanisms and colony-level outcomes.

Plant-derived acids such as p-coumaric and indole-3-acetic acid play a distinct role in detoxification. They upregulate cytochrome P450 enzymes, improving resilience to pesticide exposure and enhancing survival under acaricide stress [31,32]. These detoxification pathways are critical under environmental stress and complement the glandular enhancements described in Section 5.

Microbial interactions further amplify these effects. Acetic acid not only regulates gut pH and inhibits pathogens [76,80] but also contributes to hive hygiene and social communication, serving as a component of alarm pheromones [81]. Its role in moisture regulation and brood development underscores its multifunctionality in colony-level physiology [82,83]. While parallels with poultry and mammalian systems suggest conserved mechanisms of growth and immune modulation [63,84], insect-specific pathways, particularly in honeybees, require further validation. Nonetheless, the convergence of metabolic, microbial, and genetic evidence positions organic acids as potent modulators of honeybee physiology, bridging nutritional inputs with molecular outcomes.

## 7. Impact of Organic Acids on Honeybee Health

Organic acids play a multifaceted role in promoting honeybee health, particularly under conditions of nutritional stress, pathogen pressure, and environmental challenges. Their effects span individual physiology, glandular development, immune modulation, and colony-level resilience, making them valuable additions to supplementary feeding strategies in sustainable apiculture. Motta and Moran [75] and Elbaz et al. [84] have reviewed the health-promoting effects of organic acids, particularly their roles in microbiome stabilization, immune modulation, and detoxification. Table 2 provides an overview of colony-level health benefits linked to organic acid supplementation, including mechanisms of immunity, detoxification, and microbial support [60,70,75]. The strategic relevance of organic acids in apiculture extends beyond physiological benefits to encompass broader sustainability goals. By reducing pathogen pressure, enhancing detoxification pathways, and supporting microbial symbiosis, these compounds offer a low-residue alternative to synthetic treatments. Their integration into supplementary feeding protocols aligns with principles of integrated pest management, minimizes chemical residues in hive products, and supports regulatory compliance in honey production [31,32,81,85,86]. As such, organic acids represent a scalable, biologically grounded intervention for improving colony resilience under increasingly complex environmental stressors.

Organic acid supplementation significantly enhances gut health and nutritional physiology. Acetic and lactic acids, utilized as dietary additives, lower luminal pH and inhibit pathogenic microorganisms while promoting the proliferation of beneficial gut symbionts such as *Snodgrassella alvi* and *Gilliamella* spp. [60,70]. These acids serve as growth substrates for specialized bacteria, enabling metabolic cross-feeding and microbiome stabilization [70]. For instance, *Snodgrassella alvi* metabolizes host-derived acids such as citrate and glycerate to sustain aerobic respiration, enabling stable colonization even under carbohydrate-restricted conditions [70]. This mutualistic interaction improves nutrient absorption, ileum morphology, and digestive efficiency, contributing to worker longevity and overall colony health [60,75].

Citric acid enhances amino acid metabolism and supports HPG development by activating the TCA cycle and promoting royal jelly biosynthesis [23,71]. Supplementation increases pollen consumption and glandular activity, improving brood care and productivity [23,73]. Lactic and acetic acids, especially when combined with probiotics such as *Lactobacillus rhamnosus* or *L. brevis*, exhibit strong antimicrobial effects and bolster resistance to *Nosema ceranae* infections [33,60,64]. These acids also regulate gut microbiota, stimulate antimicrobial peptide expression, and reduce sporulation, thereby strengthening immune defenses and increasing colony vitality [76,77,87,88]. LAB-derived compounds such as phenyl-lactic acid further enhance pathogen resistance and modulate oxidative stress responses [64,85].

Through activation of antioxidant enzymes like superoxide dismutase and catalase, organic acids contribute to oxidative stress mitigation and stimulate cytochrome P450-mediated detoxification pathways [31,32,85]. Plant-derived acids like p-coumaric and indole-3-acetic acid improve survival under pesticide exposure, notably tau-fluvalinate [31]. Additionally, oxalic and formic acids exhibit acaricidal effects against *Varroa destructor*, suppressing mite populations without inducing resistance, though high dosages may transiently increase worker mortality [43,89]. Acetic acid also contributes to hive hygiene and social behavior by modulating alarm pheromones and antimicrobial surface defenses [76,81]. Collectively, these effects underscore the strategic value of organic acids in promoting resilient, productive colonies while aligning with integrated pest management and sustainable apiculture practices [85,86].

In summary, organic acids offer a multifaceted and biologically coherent strategy for enhancing honeybee health. Their roles in gut microbiome stabilization, immune modulation, detoxification, and disease resistance are supported by a growing body of evidence across nutritional and therapeutic contexts. Whether administered as standalone supplements, probiotic-acid combinations, or botanical derivatives, these compounds contribute to colony vitality, productivity, and resilience. Importantly, their use supports sustainable apiculture by reducing reliance on synthetic acaricides, lowering chemical residues in hive products, and aligning with integrated pest management frameworks [86]. As environmental pressures on pollinators intensify, the strategic deployment of organic acids may serve as a cornerstone of adaptive, evidence-based beekeeping.

**Table 2 insects-16-01203-t002:** Colony-level health benefits associated with organic acid supplementation.

Organic Acid	Supplement Type *	Health Benefit	Mechanism/Outcome	Key References
Citric Acid	Nutritional	Longevity, energy metabolism	TCA cycle activation, fatty acid synthesis, enhanced HPG development	[23,71]
Lactic Acid	Nutritional/Probiotic	Immunity, pathogen resistance	Antimicrobial peptides, reduced Nosema load, oxidative stress mitigation	[33,60,64,85]
Acetic Acid	Nutritional/Probiotic	Hive hygiene, gut health	pH regulation, symbiosis with Bombella spp., antimicrobial surface defense	[60,76,77,78]
Formic/Oxalic Acid	Non-nutritional treatment	Varroa mite control	Acaricidal effects, inhibition of oxidative phosphorylation	[43,86,89]
Host-Derived Acids	Endogenous (bee-secreted)	Microbiome support	Growth of *Snodgrassella alvi*, nutrient processing, immune pathway activation	[70,75]
Phenyl-Lactic Acid	Probiotic metabolite	Antimicrobial, detoxification	Pathogen inhibition, antioxidant enzyme activation	[64,85,87]
Plant-Derived Acids	Nutritional (botanical)	Pesticide resilience	CYP450 activation, increased survival under tau-fluvalinate exposure	[3,32]

* Supplement types were clarified to distinguish nutritional acids, probiotic-acid combinations, and non-nutritional treatments. Citations [31,32,33,81,87,88] were integrated to reflect microbial, detoxification, and behavioral effects.

## 8. Future Research Directions

While preliminary findings are promising, further research is essential to elucidate the physiological and ecological implications of organic acid supplementation on gland development, colony health, and long-term resilience. Navarro-Escalante et al. [88] advocate for microbiome engineering and integrative research approaches, aligning with calls for interdisciplinary strategies to enhance honeybee resilience. To date, most studies have emphasized short-term physiological outcomes, such as glandular activation and survival, while long-term effects on colony dynamics, reproductive success, and overwintering capacity remain underexplored. As Wright et al. [42] underscore, supplement efficacy should ideally be validated under colony-level conditions rather than caged environments, which may constrain behavioral expression and ecological interactions. Future studies should prioritize field trials that reflect real-world stressors, social dynamics, and nutritional variability. Bridging these gaps will require integrative research that combines molecular, behavioral, and ecological approaches to assess colony-level outcomes under diverse environmental conditions. Future studies should employ gene knockdown and overexpression techniques to identify regulatory pathways linking dietary acids to glandular phenotypes and metabolic performance [2,16,72,90].

Investigating how organic acids modulate microbial community structure and metabolic function may reveal mechanisms underlying gut health, immunity, and nutrient assimilation [75,76,77]. Rigorous field trials are essential to evaluate the efficacy of organic acid supplementation across varied landscapes, forage availability, and seasonal stressors, ensuring ecological validity and practical relevance [76,77]. Longitudinal studies tracking brood development, foraging behavior, and overwintering success will generate actionable data to refine region-specific feeding protocols and inform sustainable apiculture. Finally, interdisciplinary research, spanning entomology, microbiology, agronomy, and policy, will be vital for translating laboratory insights into scalable, context-sensitive feeding strategies for beekeepers [67,91]. Advancing organic acid supplementation science demands a multifaceted strategy, combining molecular insights, field validation, and stakeholder engagement, to support resilient and productive honeybee colonies.

## 9. Conclusions

While preliminary findings are promising, further research is essential to fully elucidate the implications of organic acid supplementation on honeybee gland development and overall colony health. Citric, acetic, and lactic acids demonstrate potential to enhance glandular activation, immune function, and worker longevity via metabolic and microbial pathways. These effects collectively support colony resilience under nutritional deficits and environmental stressors. Integrating organic acids into supplementary feeding strategies presents a sustainable avenue for enhancing honeybee health; however, long-term field validation and mechanistic studies remain critical to confirm efficacy and scalability. As beekeeping practices evolve under ecological pressures, organic acid supplementation may serve as a cornerstone of adaptive, evidence-based apiculture.

## Figures and Tables

**Figure 1 insects-16-01203-f001:**
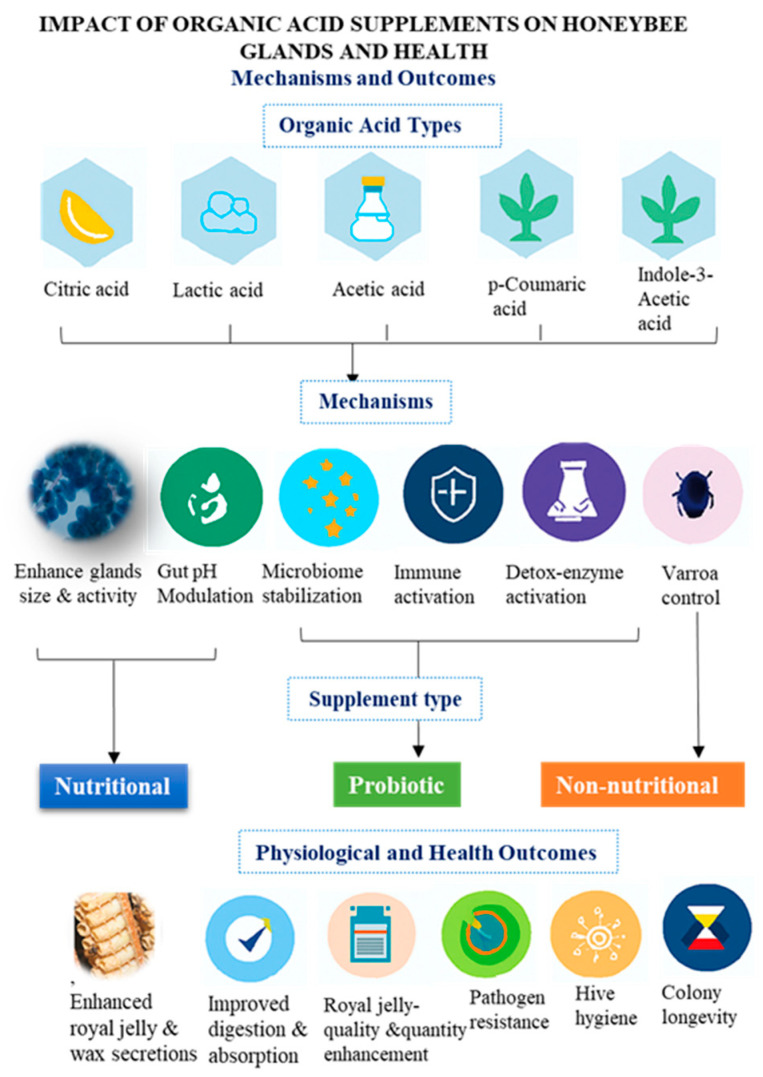
Conceptual framework linking organic acid types to physiological mechanisms and colony-level outcomes in honeybees. The diagram illustrates relationships among acid categories (citric, lactic, acetic), target mechanisms (glandular activation, gut pH modulation, immune stimulation), supplement types (nutritional, probiotic, non-nutritional), and outcomes such as enzyme synthesis, microbial balance, pathogen resistance, and colony longevity. This figure complements Table 1 and Section 5, with broader health implications discussed in Section 7.

**Table 1 insects-16-01203-t001:** Target glands and physiological effects of selected organic acids in honeybees. This table maps selected organic acid to its anatomical targets and functional outcomes, emphasizing gland morphology, biosynthetic activity, and detoxification pathways. Supplement types are categorized as direct acids (e.g., citric acid), microbial derivatives (e.g., acetic acid from *Bombella* spp.), plant-derived acids (e.g., p-coumaric acid), and probiotic-acid combinations (e.g., lactic acid with *Lactobacillus brevis*). Effects are primarily based on honeybee studies, with cross-species extrapolations noted in the text or expanded in Section 7. Dosage ranges (e.g., 0.5–0.75% citric acid) reflect experimental concentrations and may vary with colony age, diet composition, and environmental context. Abbreviations: HPG = hypopharyngeal gland; MDG = mandibular gland; CYP450 = cytochrome P450 detoxification enzymes.

Organic Acid	Supplement Type	Target Gland(s)	Physiological Effect	Key References
Citric Acid	Direct acid (synthetic/natural)	MDG, HPG, Salivary glands	Enhances gland size, royal jelly quality, wax secretion, pollen consumption	[23,71,72,73]
Lactic Acid	Microbial derivative/probiotic	HPG	Increases acinar surface area, boosts enzyme synthesis, reduces pathogen load	[33,64]
Acetic Acid	Microbial derivative (*Bombella* spp.)	Gut, Wax glands	Supports digestion, wax secretion, microbial balance, hive hygiene	[75,76,77,78]
p-Coumaric Acid	Plant-derived acid	CYP450s-mediated detoxification pathways	Upregulates CYP450s, improves pesticide tolerance	[31,32]
Indole-3-Acetic Acid	Plant-derived acid	Detox pathways	Enhances survival under acaricide stress	[31]

## Data Availability

No new data were created or analyzed in this study.

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
