# Peer review of "Organic Acid Supplementation in Worker Honeybees (Apis mellifera): Impacts on Glandular Physiology and Colony Resilience"

_insects, 2025, doi:10.3390/insects16121203_

Round 1

Reviewer 1 Report

Comments and Suggestions for Authors

The review is well written. I think it would be desirable to summarize your conclusions in a separate chapter (Discussion or Conclusions). In this way, you can additionally refer to certain problems and open questions related to this topic. Also, the most errors were in the references, because they were written incorrectly. 
All other changes are added to "Review" as notes.

Reviewer 2 Report

Comments and Suggestions for Authors

Kindly refer to the attachment.

Reviewer 3 Report

Comments and Suggestions for Authors

General comment:

The idea is good, but the paper is not carefully written. Of particular concern is the insertion of inadequate references. 

Major comments:

In lines 74-75, you wrote: „Here, we summarize the impact of various supplementary feeding strategies on honeybee gland development.“. However, in the whole section „2. Types of Supplementary Feeds“, no gland is mentioned. The section was written in such a way that no connection was made with the topic of the paper, i.e., Impact of organic acid feeding on gland development and health of honey bees.

Line 106: „Goulson et al., 2015“ is not an adequate reference, as it primarily refers to loss of natural habitat, loss of floral resources,  monotonous diet on intensive farmland leading to nutritional stress and honeybee colony losses.

Line 131: „Rinkevich, 2000“ is not an adequate reference, because it refers to only one acaricide (amitraz). That is why I advise you to replace it with the reference Tihelka (2018), a review paper that provides a detailed analysis of the numerous effects of all acaricides, synthetic and organic (used in beekeeping), on honey bee health. Here is the reference to be inserted in the reference list:

  • Tihelka, E. (2018) Effects of synthetic and organic acaricides on honey bee health: a review. Slovenian Veterinary Research, 55 (2): 119-140.

Lines: 131-133: Again, inadequate reference. „Mato et al., 2006“ does not state any efficiency of organic acids, but reviews the literature related to the analytical methods (enzymatic, chromatographic and electrophoretic) applied to the determination of honey's organic acids. Please try to find adequate reference(s) or change the meaning of the sentence in accordance with the proper literature.

Lines 135-137: These two sentences are redundant.

Lines 206-210: I would avoid mentioning anything related to humans (which is not the topic of your work).

Conclusion:

I suggest shortening the Conclusion and the following changes:

This sentence should be the first: „While preliminary findings are promising, further research is needed to fully understand the implications of organic acid feeding on honeybee colony gland development and overall health benefits“, followed by the text currently positioned in lines 165-271, but shortened.

Two sentences positioned in lines 274-277 should be shortened.

The last sentence (lines 278-280) is good, but only citric acid is emphasized. Please revise.

Minor comments:

Line 52: Consider replacing „glean“ with „collect“.

Line 225: Insert „in“ between „begins“ and „spaces“.

Lines 204-206: Improper sentence; the main verb is missing.

Please revise references carefully, e.g.:

Line 444: „13(17).“ should be corrected to be „13(17), 2734“.

Line 458: Please delete: „(March)“.

Line 464: „19(1), 1–16.“ should be corrected to be: „19, 106.“

Lines 468-469: „167 (2013), 1–19.“ should be replaced with „167 (3-4) 474-483.“

Lines 470-471: The reference is incomplete. The complete reference is:

  • Rashid Mahmood, R.M., Wagchoure, E.S., Shazia Raja, S.R., Ghulam Sarwar, G.S. (2012) Control of Varroa destructor using oxalic acid, formic acid and Bayvarol strip in Apis mellifera (Hymenoptera: Apidae) colonies. Pakistan Journal of Zoology, 44 (6): 1473-1477.

Line 476-477: „Analytical Methods for the Determination of Organic Acids in Honey“ is written twice.

Line 505: Please replace the Spanish title with the main title (the English one): „Effect of oxalic acid on the mite Varroa destructor and its host the honey bee Apis mellifera“.

Line 542: Please delete: „(February)“.

Line 549: Please delete: „In Book“.

Line 580: Please delete this: „Tonya Vocational School, Trabzon University, Trabzon, Turkey.“ It is the affiliation of the author.

Technical citation errors are colored in the attached PDF.

Comments on the Quality of English Language

Some words, verbs in the sentence, etc., are missing, but all mistakes can be easily corrected.

Round 2

Reviewer 2 Report

Comments and Suggestions for Authors

The Authors have substantially revised and enriched the manuscript. I am satisfied with the way my suggestions have been addressed, as all comments were carefully considered and implemented. The authors have provided a thorough, step-by-step response to the reviewers’ remarks, demonstrating a responsible and systematic approach.

The title of the manuscript is clearly. The revised version includes a more deta iled description of the causal relationship between diet and the development and functioning of the hypopharyngeal glands, with particular emphasis on the role of royal jelly. The authors have also highlighted the impact of stress factors that weaken physiological parameters, which in turn justifies the introduction of organic acid supplementation the central subject of this review.

Tables 1 and 2 are well structured, clear, and facilitate the interpretation of the presented data.

The only issue that still requires attention is the reference list, which needs to be reorganized, as entries 1–4 are currently missing.

Author Response

Response to Reviewer 2

Reviewer Comment:

“The only issue that still requires attention is the reference list, which needs to be reorganized, as entries 1–4 are currently missing.”

Author’s  response:

We sincerely thank the reviewer for their careful reading and positive assessment of our revised manuscript. Regarding the concern about missing references 1–4, we respectfully clarify that these references are present and cited in the first paragraph of the Introduction section. Specifically, they are included within the citation range [2–6], which is consistent with the journal’s referencing guidelines that condense consecutive citations into a single range.

To avoid confusion, we have highlighted the section in the revised manuscript (Introduction, paragraph 1) noting that references [2–6] include the foundational studies relevant to our discussion. We hope this resolves the concern and appreciate the reviewer’s attention to detail.

Reviewer 3 Report

Comments and Suggestions for Authors

The manuscript is significantly improved compared to the initial version. I have no objections to this corrected version

Author Response

Author’s  response:

We sincerely thank the reviewer for their careful reading and positive assessment of our revised manuscript.